# Feline Forensics: Revealing the Unique Decomposition of Cats

**DOI:** 10.3390/ani14070996

**Published:** 2024-03-24

**Authors:** Kelly Bagsby, Adam W. Stern, Krystal R. Hans

**Affiliations:** 1Department of Entomology, Purdue University, 901 W State St., West Lafayette, IN 47907, USA; hans3@purdue.edu; 2College of Veterinary Medicine and College of Medicine, University of Florida, Gainesville, FL 32611, USA; adamstern@ufl.edu

**Keywords:** veterinary, TBS, pathology, animal cruelty

## Abstract

**Simple Summary:**

This study analyzed the decomposition of cats, identified any unique decomposition characteristics, and determined the rate of decomposition using total body score (TBS) and accumulated degree days (ADDs). Determining the stages of decomposition is difficult because the stages do not have discrete beginnings and ends. Since there are so many different species of animals in the world that have very different characteristics, such as fur, size, and body fat, this adds another layer of difficulty to the stage of decomposition determination. Overall, there was not a significant difference between the rate of decomposition between light- and dark-furred cats. Unique decomposition characteristics were observed, such as fur obscuring details of decomposition, among others. Understanding these unique findings has considerable implications in future investigations involving the neglect and abuse of cats. Applying the findings would assist pathologists and veterinarians in their determination of neglect and abuse timeframes.

**Abstract:**

Limited data about the differences between the decomposition of animals with fur exist within the forensic veterinary medicine discipline. Due to the sheer number of animals used as animal models and the number of animals that exist, determining stages of decomposition that are applicable for all species is difficult. Typically, assessing what stage of decomposition a decedent is in is subjective due to the visual assessment of decomposition changes. A method developed to rectify this issue was the total body score (TBS) method, which assigns a numerical value based on the degree of decomposition to the head, torso, and limbs. The purpose of this study was to examine the decomposition of cats, identify any unique decomposition characteristics, and determine the rate of decomposition using total body score and accumulated degree days (ADDs). Twelve domestic short-haired cats were placed in a grassy field in West Lafayette, IN. An assessment of TBS was documented for each cat and each experimental group. An exponential relationship between TBS and ADD was documented. Overall, there was not a significant difference in the rate of decomposition or total body scores between the groups (Z = −91.00, *p* = 0.0672).

## 1. Introduction

Veterinary forensic medicine is a novel discipline within the veterinary medicine and forensic science fields. Many subdisciplines constitute forensic veterinary medicine, such as pathology, toxicology, wildlife conservation, and more. An integral aspect of this field is veterinary forensic pathology. This subdiscipline is responsible for identifying any injuries or disease, documenting and interpreting these medical findings, and finally relaying the evidence to judicial parties [1]. Interpreting the evidence found through medical examinations is critical for the judicial process so that judges, attorneys, and the jury can understand the evidence, the severity of injuries or diseases, and the implications of such evidence. The responsibilities of a veterinarian will vary by specialty. Some of the responsibilities include, but are not limited to, triaging; conducting live examinations; providing appropriate treatments; performing autopsies/necropsies; identifying and documenting the scene, evidence, and medical findings; providing expert testimony [2]. Involvement by veterinarians, especially veterinary pathologists, is critical in the investigations of animal cruelty, abuse, or neglect and any post-mortem examinations, if warranted. Currently, all fifty states in the United State consider animal cruelty to be a felony [3].

Post-mortem examinations provide veterinarians with crucial information about how long an animal suffered, the cause of death, and how long an animal has been deceased [4]. During these investigations, law enforcement agencies may request a timeframe of abuse or time since death, which is known as the post-mortem interval (PMI). Estimating PMI is difficult and there are many different methods used, including forensic entomology and total body score [5]. Animals undergo the same stages of decomposition as humans but there are differences [6,7]. Temperature-based estimations of the rate of body cooling are widely used, but their accuracy is low and extrapolation of human data for use on animals does not comprise consistent results [6]. Estimating the PMI for animals using forensic entomology has also shown differences compared to human studies. Forensic entomology is the study of insects, wherein entomologists answer questions in the judicial system [8]. There is an added layer of difficulty when attempting to determine the later stages of decomposition. The morphological differences are subtle in the later stages once fly larvae migrate away from the remains. Other methods to determine advanced PMIs include radiocarbon/radioisotope dating, alternative light source techniques, bone composition, and proteomics [9].

The collection, rearing, and identification of blow flies eggs can determine the time of colonization. The time between when blow flies lay their eggs on the remains, subsequently feed, and when the remains are discovered is the time of colonization (TOC) [10]. Immature insect samples (eggs, larvae, and pupae) are collected in duplicate, with a preserved sample and a reared sample. The preserved sample is collected to prohibit further development of the insects, which allows a forensic entomologist to determine the stage of insect when collected form the remains. The reared sample is collected to raise the insects to the adult stage for ease of identification [11]. Research has shown that certain species develop at different rates and have different minimum and maximum threshold temperatures. Based on this species identification, a forensic entomologist can estimate when the remains were colonized by insects, which aids in determining a time frame for death investigations.

Another method for determining PMI is total body score (TBS), which is a way to quantitatively determine the stage of decomposition [7,12]. An influential anecdotal study assigned TBS to human remains when the day of death was known [13]. In this study, TBS was associated with accumulated degree days (ADDs) with the goal of determining the PMI. When using the TBS method, three different regions of the body are assigned a numerical value based on descriptions of the degree of decomposition. The head is scored from 1, fresh, to 13, dry bone; the torso is scored from 1, fresh, to 12, dry bone; and the limbs are scored from 1, fresh, to 10, dry bone. As decomposition progresses, morphological changes, such as skin slippage, bloating, and moist decomposition, are observed with greater scores being assigned, which increases TBS. Many studies have used TBS to determine the PMI and, in general, there are many shortcomings. A universal equation that is applicable to all animal species is unattainable with current resources [12,13,14,15]. Accumulated degree days (ADDs), a measurement of the sum of the average daily temperatures, can be associated with TBS to determine the rate of decomposition.

Another area of contention is defining the stages of decomposition of furred animals, durations of each stage, or terms and definitions of each stage (Table 1). This difficulty stems from the continuous process of decomposition with little to no defined boundaries of when one stage begins and ends. Assessment of each stage also depends upon visual assessments that are subjective to the observer [16]. Decomposition begins the moment that death occurs, with autolysis and putrefaction occurring immediately [17]. The stages of decomposition described here are fresh, bloat, active decay, advanced decay, and dry [17,18,19]. The fresh stage can be described as no discoloration and no insect activity [17]. Bloat can be described as when the body begins to decompose and microorganisms proliferate, which produces gases such as methane and ammonia; these gases then inflate the abdominal cavity, giving a bloated appearance [19,20]. Active decay can be described as moist decomposition with extensive insect activity, and the body may have a deflated appearance now due to insect feeding breaking the skin layers [17]. During this stage, the temperature of the carcass peaks [19]. Advanced decay is continued moist decomposition with some bone exposure, transitioning to a dryer mummified stage [12]. Finally, there is the dry stage, when there is only dry skin and no insect activity [19,21]. Visual assessment of the morphological changes introduces additional difficulty due to fur concealing the surface of the skin. Many research studies have used pigs because they have the most similar decomposition to human [22]. Few studies have used cats, or even animals with fur, as the animal model when analyzing decomposition [7,19,21,23]. When assessing the degree of decomposition, whether the remains are human or animal, the process is very subjective to the observer and may result in different conclusions. A way to remedy this is to implement the TBS method, which has been demonstrated to show low levels of any interobserver bias [16].

The purpose of this study was to examine the decomposition of cats (*Felis catus*), identify any unique morphological changes, and determine the rate of decomposition using the TBS method with ADD. The implications of these findings will aid forensic veterinary pathologists in their determination of PMI.

## 2. Materials and Methods

This study was conducted in August 2021 for a total of twelve days in West Lafayette, IN, USA, using twelve domestic short-haired cat carcasses. The cats were ethically sourced from a local animal shelter after chemical euthanasia via an intra-cardiac stick with pentobarbital sodium and phenytoin sodium. The cats were euthanized according to the shelter’s protocol and were not specifically requested for this study. Documentation included fur color and weight, with a mean weight of 5.3 +/− 0.11 kg; there were nine cats with dark fur and three cats with light fur. The cats were photographed and then placed in left lateral recumbency in a grassy field at the same time. A wire cage (1.52 m × 0.91 m × 0.61 m) was placed over each cat in order to prevent scavengers from consuming any tissues.

After the cats were placed in the field, documentation occurred every two hours on the initial day. Each observation consisted of recording the carcass temperature, ground temperature, and morphological changes. The following days, the cats were checked twice a day until there was minimal insect activity. Once they reached the dry stage, they were checked once a day until completion of this study. Cats were split into two experimental groups: light fur and dark fur. Total body scores were assessed by using the TBS chart from the Sutton 2017 study [12]. Each cat was photographed, observed to document morphological changes, and quantified based on the decomposition of each region to determine the TBS. When assessing the morphological changes, the three regions, the head/neck, torso, and limbs, were scored independently. Using the TBS chart, each region was observed and photographed. As decomposition progressed, observable changes were recorded and a score was assigned. The three regions were then summed, resulting in the total body score. The mean TBS was calculated for both light- and dark-fur experimental groups.

Field temperature and relative humidity were collected via four data loggers (HOBO MX2300, Onset Computer Corporation, Bourne, MA, USA). Body temperatures and ground temperatures were recorded daily throughout the study via an infrared digital thermometer (Infrared Thermometer GM321, Benetech, China). In order to calculate the accumulated degree days, the field temperature data collected by data loggers were corrected by using the nearest certified weather station (GHCND:USW00014835) to account for any variability [29]. Daily average field temperatures were collected and plotted against the certified weather station daily averages, and the resulting equation was used to correct the field temperature (corrected field temperature = 1.2044x − 0.9103) [29,30,31]. This equation was used for each daily mean temperature to calculate a corrected field temperature value. This process was repeated for each day of the study. The values were calculated and then summed to calculate ADD (Table 2).

### Statistical Analysis

Analyses examined the effect of fur color on TBS. The data were not normally distributed and Wilcoxon and Spearman correlation tests were conducted to compare the TBS of both light-fur and dark-fur cats. The TBS values for light- and dark-fur cats (x-axis) were plotted against ADD (y-axis) to determine the rate of decomposition. Spearman correlation and nonlinear regression of TBS and ADD were assessed for both light- and dark-fur experimental groups. The analyses also examined the effect of fur color on the body temperature of the cats. Body temperature and field ambient temperature were also assessed for both light- and dark-fur groups. Temperature data were normally distributed (Kolmogorov–Smirnov, *p* > 0.05), and a paired sample *t*-test was performed to compare body temperature for light- and dark-fur cats. A paired sampled *t*-test was also performed to analyze mean body and field temperatures. All statistical analyses were performed using GraphPad Prism version 9.5.0 for Windows (GraphPad Software, San Diego, CA, USA, www.graphpad.com (accessed on 8 December 2022).

## 3. Results

Total body scores were not significantly different between the light- and dark-fur cats (Z = −91.00, *p* = 0.067) (Table 2). There was a strong positive correlation between the total body scores of light- and dark-fur cats though, which was statistically significant (r(10) = 0.984, *p* = 0.0001) (Figure 1).

The rates of decomposition for both light- and dark-fur cats had an exponential relationship with high R^2^ values, respectively (R^2^ = 0.84, R^2^ = 0.90) (Figure 2 and Figure 3). Light-fur TBS vs. ADD was significantly different (Z = 78, *p* = 0.0005). Dark-fur TBS vs. ADD was also significantly different (Z = 78, *p* = 0.0005). 

The maximum body temperatures for light- and dark-furred cats were 39.18 °C +/− 2.2 and 41.91 °C +/− 3.3, respectively. The minimum body temperatures for light- and dark-furred cats were 21.30 °C +/− 1.0 and 25.39 °C +/− 1.8, respectively. The maximum relative humidity was 82.74%, and the minimum was 52.17%. Although body temperatures between the experimental groups were significantly different (t = 3.925, df = 11, *p* = 0.0024), there was no significant difference between the mean body temperatures and mean field temperatures (light fur: t = 0.8738, df = 12, *p* = 0.40; dark fur: t = 0.4184, df = 12, *p* = 0.68).

Based on the observations of the morphological changes that occurred during the decomposition processes, the decomposition stage for each cat was assessed (Figure 4). On the initial day, the cats were considered fresh with a TBS score of 3. At around hour 4–6 on the initial day, the cats transitioned into the early decomposition stage, with a score between 4 and 11. The morphological changes that were observed during this stage include pink/white discoloration, skin slippage, slight insect activity, purge fluid, and drying of the nose/mouth. Bloat was observed as early as hour 8 on the initial day and continued until day 2. During in the bloat stage, TBS was scored between 12 and 17. Feeding by insects was prominent as early as day 1, and the larvae continued to voraciously feed until day 4. Insect activity was documented to occur from the initial day. Active insect feeding was prominent, as evident by the fur displacement, but was obscured due to the fur. This early stage of insect feeding coincided with the early decomposition and bloat stage. When larvae fed to the point that the skin was broken and post-bloating occurred, some tissues began to discolor to a black/brown. The active decay stage of decomposition occurred during day 1–4, with moist decomposition being actively observed where the TBS was scored between 6 and 21. Once the larvae were finished feeding on around day 4–5, they migrated away. This marked the transition period between the active stage and the advanced stage, when the abdomen and thoracic cavities caved in and tissues began to dry out. TBS during this advanced stage was between 19 and 24. As the tissues continued to dry out and become mummified during days 5–11, the TBS increased slightly from 24 to 27, which was considered the dry stage. Few cats reached the skeletonization stage, associated with a TBS of 28+, but those that did showed considerable bone exposure, with whole limbs or the ribcage being readily visible. Partial skeletonization in some cats occurred as early as day 4.

## 4. Discussion

This study provides information about the decomposition of cats with regard to light and dark fur. The similarities between the TBS scores show that fur color did not impact the progression through decomposition. Dark-fur cats did appear to reach the skeletonization stage more often than the light-fur cats did. The maximum TBS that light cats reached was 25, but the dark-fur cats reached a maximum of 29. The degree of decomposition reaching skeletonization could have been due to temperature, insect activity, internal variables such as diet and microbial activity, or other external environmental variables, but more research needs to be conducted in order to analyze why the dark cats reached skeletonization.

Fur color did affect the body temperature of the cats. Dark-fur cats had a significantly higher body temperature compared to light fur cats. Dark-fur cats had an average max body temperature of 41.9 °C on day 3, with one cat reaching a maximum of 54 °C. The reason that the temperature was higher in the dark-furred cats could be due to the absorption of heat from the environment and solar radiation [32]. There are many environmental factors that affect the rate of decomposition, and research has shown that temperature is one of the most important factors [6], as seen by the high R^2^ values when analyzing TBS vs. ADD. Since body temperatures and field temperatures were not significantly different, these findings reinforce the concept that temperature affects the rate of decomposition. Other factors include insect activity, coverings such as clothes or fur, trauma, and more [6]. 

Other variables that were not analyzed in this study may affect TBS, and these could be assessed in the future. Some cats had very unique morphological findings. The fur of cat 10 was never displaced, even on day 6, but the insect larvae were feeding so there was little-to-no observable skin discoloration, especially when trying to assess the score for the thorax (Figure 5a,b). Fur displacement occurred for almost all of the cats, resulting in a canopy-like covering, which was also observed the in later morphological changes such as mummification and bone exposure (Figure 5c,d). The progression through decomposition also varied within each experimental group. Cat 4, a light-fur cat, progressed to the advanced stage rapidly on day 4, showing mummification and fur displacement, compared to other cats (days 5–11) (Figure 5d). Cats 6 and 11 had observable differences on the same day of the study, where cat 6 was in the bloated stage and cat 11 was partially skeletonized (Figure 5e,f). Future studies could analyze the physiology of each animal, such as biomass, nutritional state, and medical history. The physiological differences could influence the way decomposition progresses, the observable morphological changes, and TBS score. 

Understanding how animals progress through the stages of decomposition is important for the veterinary and forensic science fields. There have not been many studies analyzing the decomposition of animals with fur [7,19,21]. Of the studies analyzing animals with fur, assessment of TBS was impacted due inability to observe color changes, skin slippage, and bloat [7]. Early and Goff (1986) have analyzed the decomposition of cats and determined that there were five stages of decomposition: fresh, bloated, decay, dry, and remains [19]. The present study has found similar results as the 1986 study, where the initial day was the fresh stage, the end of the initial day consisted of early decomposition, days 0–2 were bloat, days 1–4 were active decay, days 4–5 were advanced decay, and days 5–11 were the dry stage. Also, the cats’ progression of decomposition did not follow the TBS chart fluidly.

Given these findings, a TBS chart specifically for animals with fur would be beneficial. When assessing the remains of animals with fur, color changes in the paw pads, mucus membranes of the mouth, and ear pinnae should be observed because of the lack of fur in those areas. Although, these areas do not address the area of the torso. In this study, these areas were observed, but the color changes were not as excepted according to the TBS chart. Future directions could include analyzing these areas for specific morphological changes so that TBS can be assessed accurately. The characteristics of fur should be analyzed in order to score the torso, and suggestions include disturbance, hair loss, and greasy appearance. An increase in these characteristics should align with an increase in scoring. The incorporation of insects into a scoring system would also benefit the TBS system. As decomposition progressed and insects fed, the fur was disturbed, which resulted in it being sloughed off or a mat forming, which covered the moist decomposition and skeletal remains. Gauging the degree of fur sloughing or matting could aid in assessing the TBS, especially for the torso which may not have any skin visibly available for color assessments. Decomposition resulting in sagging flesh, caving in of the abdominal/thoracic cavities, and moist decomposition seemed to occur simultaneously. In the TBS chart, these morphological features have different scores.

This study contributes information to the forensic and veterinary disciplines regarding the decomposition of animals with fur. This study shows that there are important findings that seem to be unique to cats and possibly other animals with fur. Understanding these unique findings has considerable implications in future investigations involving the neglect and abuse of cats. Assessment of the degree of decomposition can aid in determining the PMI. Future directions should include developing a TBS specific to animals with fur, understanding the minute characteristics that occur during the early decomposition stage which are obscured by fur, and assessing the effect that fur may or may not have on the rate of decomposition.

## 5. Conclusions

The field of veterinary forensics is a rapidly growing discipline. With more awareness about cruelty and an increase in animal laws around the country, more research needs to be conducted to strengthen the understanding of the physiological effects of cruelty. Understanding how animals decompose, the stages of decomposition they go through, and the unique morphological changes that occur is pertinent to determining the postmortem interval. Animals have several different characteristics that will affect the decomposition of an animal, including the presence of fur, size, and body fat stores [7,28,33,34]. Given these differences between animals and humans, this limits the extrapolation of the decomposition process between species. Therefore, the veterinary community cannot rely solely on studies of humans or pigs to determine postmortem changes in other species of veterinary and legal interests.

In this study, we saw firsthand how it was difficult to use the TBS chart developed for humans, as the cats’ fur proved to hinder our ability to detect changes in skin coloration and the presence of skin slippage. We also saw that fur color does affect body temperature but does not affect the rate of decomposition. A suggestion of how decomposition progressed was also proposed: the cats progressed through decomposition in the order of fresh, early decomposition, bloat, active decay, advanced decay, dry, and finally skeletonization. This proposal combines what has been established in the literature and what was visually assessed in this field study pertaining to furred animals. Additional studies are warranted to further study TBS in cats, and these studies should include different locations and seasons as temperature can have a major effect on the rate of decomposition.

## Figures and Tables

**Figure 1 animals-14-00996-f001:**
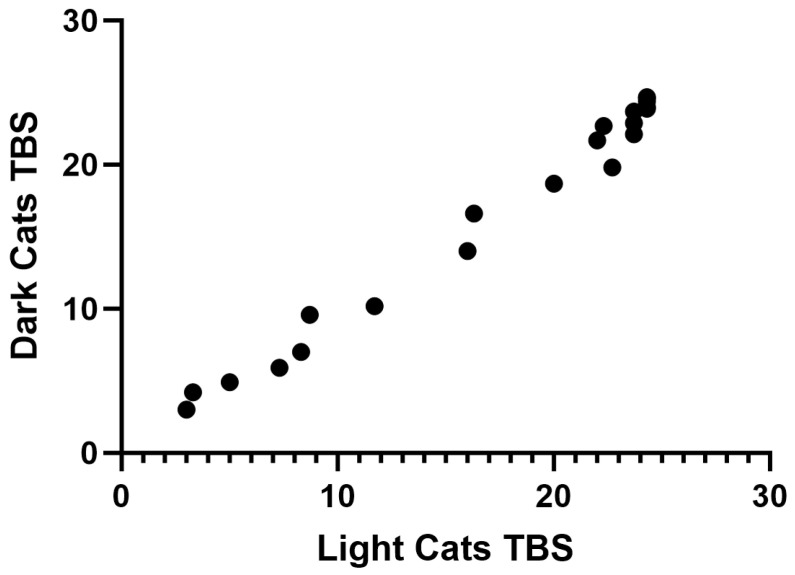
Relationship between light- and dark-fur cats had a strong, positive correlation (r(10) = 0.9843, *p* = 0.0001). As the cats decomposed, their total body scores increased similarly.

**Figure 2 animals-14-00996-f002:**
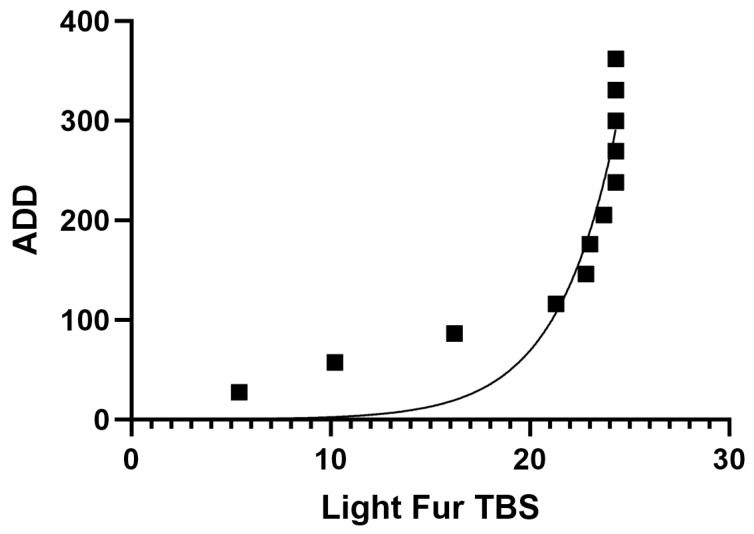
Nonlinear regression for light fur cats (*n* = 3) TBS vs. ADD resulted in an exponential relationship (R^2^ = 0.84). Light-fur TBS vs. ADD was significantly different (Z = 78, *p* = 0.0005).

**Figure 3 animals-14-00996-f003:**
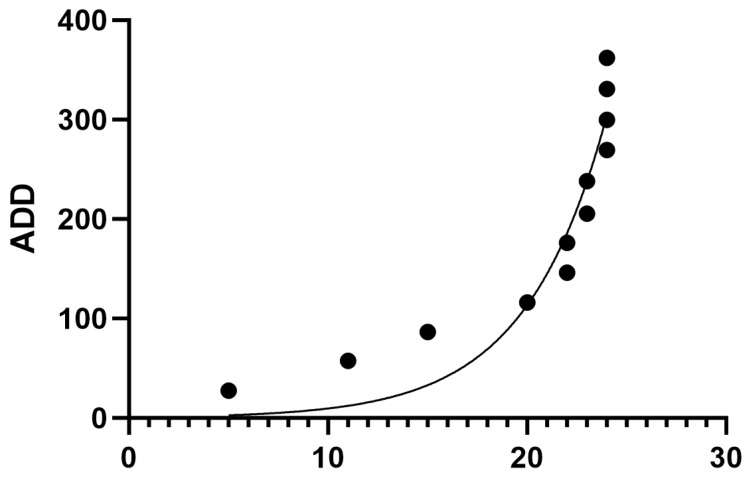
Nonlinear regression for dark-fur cats (*n* = 9) TBS vs. ADD resulted in an exponential relationship (R^2^ = 0.90). Dark-fur TBS vs. ADD was also significantly different (Z = 78, *p* = 0.0005).

**Figure 4 animals-14-00996-f004:**
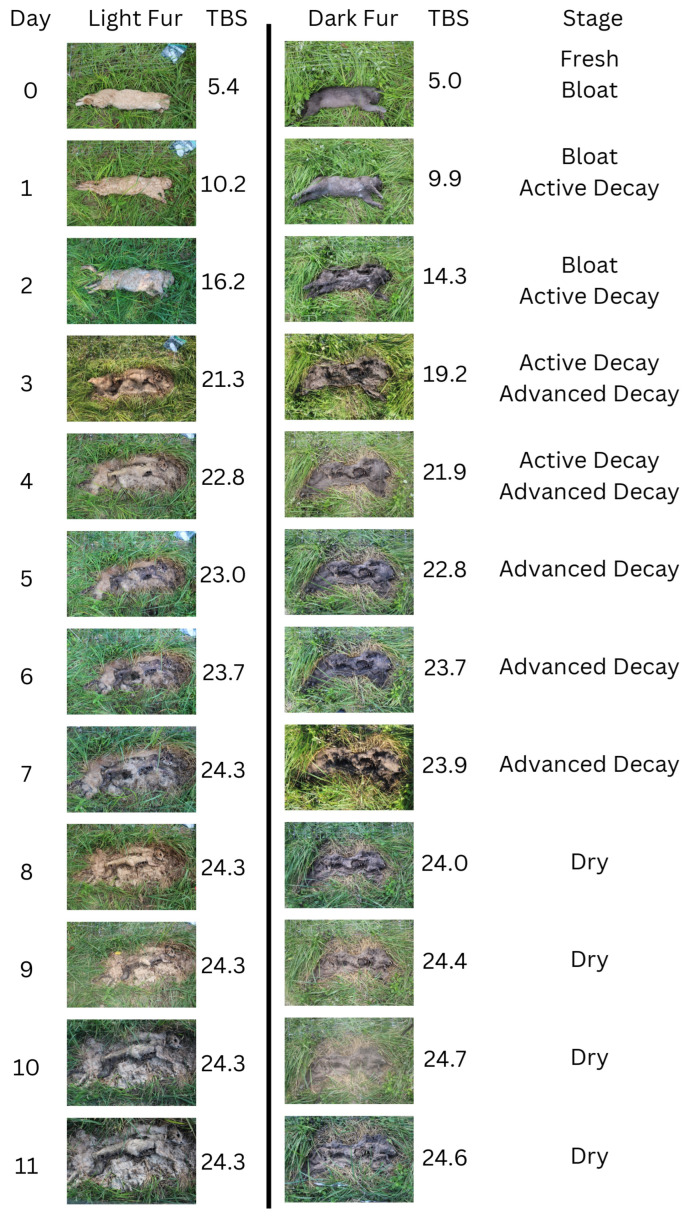
Progression through decomposition for light- and dark-fur cats depicting the morphological changes observed according to the TBS of the head, thorax, and limbs. Determination of the stage of decomposition the cats were in was a comprehensive assessment based on the average TBS score for light- and dark-fur cats.

**Figure 5 animals-14-00996-f005:**
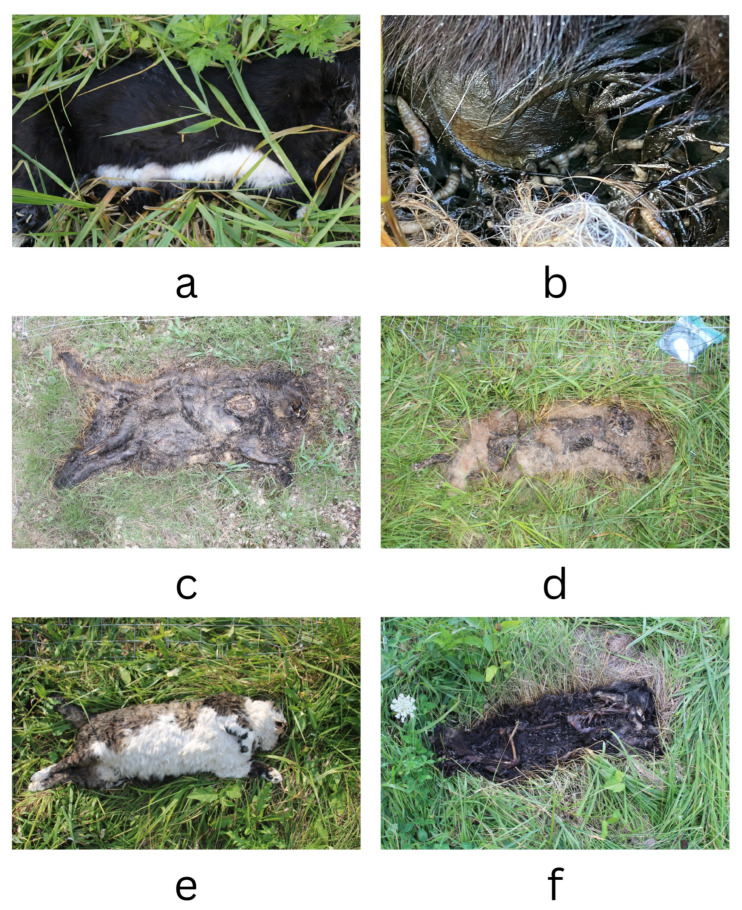
(**a**) Cat 10, day six, on the surface there is little-to-no evidence of decomposition. (**b**) Under the fur of Cat 10, there was significant decomposition, with extensive insect activity and moist decomposition. (**c**) Cat 2, day nine, a canopy-like covering of matted fur obscured any morphological changes. (**d**) Cat 4, day four, with fur displacement, some bone exposure, and mummified tissues. (**e**) Cat 6, day three, during the bloat stage. (**f**) Cat 11, day three, with extensive bone exposure.

**Table 1 animals-14-00996-t001:** Stages of decomposition presented in the literature. The number of stages range from three to six, but the literature has listed up to nine stages of decomposition.

Author	Animal	Number of Defined Stages of Decomposition
Bornemissza, 1956 [20]	Guinea pig	5
Reed 1958 [21]	Dog	4
Payne 1965 [24]	Pig	6
Early and Goff, 1986 [19]	Cat	5
Tomberlin and Adler, 1998 [25]	Rat	3
Gonder, 2008 [26]	Wolf	5
Bachman and Simmons, 2010 [27]	Rabbit	3
^1^ Brooks, 2018 [17]	-	5
^1^ Smith-Blackmore, 2023 [28]	-	6

^1^ Animal model was not provided in the text.

**Table 2 animals-14-00996-t002:** Temperature and TBS raw data for the duration of the study. To calculate ADD, the thermal units for each day were summed. TBS was not significantly different between light and dark fur (Z = −91.00, *p* = 0.067), and TBS vs. ADD for both groups resulted in an exponential relationship (R^2^ = 0.84, R^2^ = 0.90).

Day	ADD	Light-Fur TBS	Dark-Fur TBS
0	27.53	5.4	5.0
1	57.49	10.2	9.9
2	86.60	16.2	14.3
3	116.14	21.3	19.2
4	146.01	22.8	21.9
5	176.21	23.0	22.8
6	205.41	23.7	23.7
7	238.29	24.3	23.9
8	269.71	24.3	24.0
9	299.71	24.3	24.4
10	330.92	24.3	24.7
11	362.13	24.3	24.6

## Data Availability

Data is publicly available at the following link: https://doi.org/10.6084/m9.figshare.25466362 (accessed on 19 March 2024).

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
