# Peer review of "Feline Forensics: Revealing the Unique Decomposition of Cats"

_animals, 2024, doi:10.3390/ani14070996_

Round 1

Reviewer 1 Report

Comments and Suggestions for Authors

The authors present here a lovely and interesting manuscript of high relevance in the field of veterinary forensic investigations, finally aiming at producing data that can be used in the field by both pathologists and law enforcement agencies.

Minor Comments

·         Line: the word “treatment” in the abstract is misleading term, both because it is not anticipated or described in the abstract and because later in the text the word treatment is used to indicate the two experimental groups.  

·         Line 64: the capital “I” should be removed.

·         Line 67: the explanation about the methodology for forensic entomology is incomplete. The options I would suggest are either remaining more vague or explaining instead in full detail since PMI can be esteemed from the “fixed” fly larvae without the need to let them rear to adult for definitive species recognition.

·         Line 106: “…when the gases within the body begin to decompose themselves….”. can gas decompose or maybe you intend when gases produced during putrefaction seep through decomposed tissues and…

·         Line 116: incomplete sentence: “An additional issue that arises when determining stage of decomposition of a furred animal”. Maybe removing “that” will make the sentence stand autonomously.

·         Line 187: the writing of the current paragraph is difficult to follow. Light fur TBS vs ADD has a strong correlation…then in line 190 the same statement is repeated with different values.

·         How are the temperature values expressed? Are they Celsius degrees? oC should be added in compliance with international standards of nomenclature.

·         Line 213: incomplete sentence.

·         Line 228: “people”?. Please correct.

·         Line 269: most instead of more.

·         Line 290: remove the comma between early and bloat

·         Line 361: “…to determine postmortem changes…”, consider maybe adding: “..in other species of veterinary and legal interest”

·         Line 362: remove “show”.

·         Images: be sure that the dimension of the photograph is consistent, since in the Fig.4 many photos have smaller dimensions.

Other Comments

·         Were cats placed on the ground all at the same time?

·         How long was the duration of the experiment, in days? This info is no very clearly stated in the first part of the M&M paragraph, where instead the impression is given to the reader that the choice of experimental length is reasoned by the achievement of the skeletonization stage of decomposition rather than a consistent length of time.

·         Fig. 6 shows two cats at day 3, one skeletonised while the other in bloating stage. It would be important to know if there is a link with the nutritional state (obese VS normal or thin) and to state clearly if the animals have been laid on the ground in different moments. This is mentioned in line 318 but no further comments are made aimed at explaining. What was their difference in ADD?

Comments on the Quality of English Language

Apart from the presence of few scattered sentences with incorrect synthaxis or typos, no further comments can be made with regard to the use of written English.

Author Response

Thank you to the reviewer for your thoughtful suggestions and feedback. We have made the changes requested and have listed the changes below:

Line: the word “treatment” in the abstract is misleading term, both because it is not anticipated or described in the abstract and because later in the text the word treatment is used to indicate the two experimental groups.  

Changed all mentions of “treatment” to “experimental groups” or “groups”

Line 64: the capital “I” should be removed.

Changed to lowercase

Line 67: the explanation about the methodology for forensic entomology is incomplete. The options I would suggest are either remaining more vague or explaining instead in full detail since PMI can be esteemed from the “fixed” fly larvae without the need to let them rear to adult for definitive species recognition.

Added/reworded section to include information about paired samples: “Immature insect samples (eggs, larvae, and pupae) are collected in duplicate, with a preserved sample and a reared sample. The preserved sample is collected to prohibit further development of the insects, which allows a forensic entomologist to determine the stage of insect when collected form the remains. The reared sample is collected to raise the insects to the adult stage, for ease of identification.”

Line 106: “…when the gases within the body begin to decompose themselves….”. can gas decompose or maybe you intend when gases produced during putrefaction seep through decomposed tissues and…

Reworded sentence to “Bloat can be described as the body begins to decompose microorganisms proliferate which produce gases, such as methane and ammonia, these gases then inflate the abdominal cavity giving a bloated appearance”

Line 116: incomplete sentence: “An additional issue that arises when determining stage of decomposition of a furred animal”. Maybe removing “that” will make the sentence stand autonomously.

Deleted “that”.

Line 187: the writing of the current paragraph is difficult to follow. Light fur TBS vs ADD has a strong correlation…then in line 190 the same statement is repeated with different values.

This was a typo, deleted these sentences to be more concise.

How are the temperature values expressed? Are they Celsius degrees? oC should be added in compliance with international standards of nomenclature.

Added .

Line 213: incomplete sentence.

Combined sentences: “On the initial day, the cats were considered fresh with a TBS score of 3.”

Line 228: “people”?. Please correct.

This was a typo, corrected to say “period”.

Line 269: most instead of more.

Changed “more” to “most”.

Line 290: remove the comma between early and bloat

Removed “early” to stay consistent with terminology used: fresh, bloat, active decay, advanced decay, and dry/skeletonization.

Line 361: “…to determine postmortem changes…”, consider maybe adding: “..in other species of veterinary and legal interest”

Added “in other species of veterinary and legal interest”.

Line 362: remove “show”.

Removed “show”

Images: be sure that the dimension of the photograph is consistent, since in the Fig.4 many photos have smaller dimensions.

Redid the figures so that all images are of the same dimensions.

Other Comments

Were cats placed on the ground all at the same time?

Yes, added to materials and methods. “The cats were photographed and then laid on their left side in a grassy field at the same time.”

How long was the duration of the experiment, in days? This info is no very clearly stated in the first part of the M&M paragraph, where instead the impression is given to the reader that the choice of experimental length is reasoned by the achievement of the skeletonization stage of decomposition rather than a consistent length of time.

Added to the materials and methods: “This study was conducted in August 2021, for a total of twelve days, in West Lafayette, Indiana, USA using twelve domestic short haired cat carcasses.”

Fig. 6 shows two cats at day 3, one skeletonised while the other in bloating stage. It would be important to know if there is a link with the nutritional state (obese VS normal or thin) and to state clearly if the animals have been laid on the ground in different moments. This is mentioned in line 318 but no further comments are made aimed at explaining.

Since this study did not analyze biomass loss, we are unable to state whether there is a link with nutritional stage and decomposition (TBS, rate, etc). Additionally, there was limited medical history information available for the cats. Added this to be a future study: “Future studies could analyze the physiology of each animal, such as biomass, nutritional state, and medical history. The physiological differences could influence the way de-composition progresses, observable morphological changes, and TBS.”

Added information about when the cats were laid on the ground: “The cats were photographed and then laid on their left side in a grassy field at the same time.”

What was their difference in ADD?

ADD was the same for all cats throughout the duration of the study since ADD was an accumulation of thermal units.

Thank you for your thoughtful consideration,

Kelly Bagsby

Reviewer 2 Report

Comments and Suggestions for Authors

The content of this article is undeniably positive in its significance for forensic veterinary medicine and animal welfare. A robust development in forensic veterinary medicine could indeed deter incidents of animal abuse. However, I find the experimental content of the article somewhat lacking in richness. Although it focuses on the stages of corpse decomposition, the content appears thin and may not be suitable for publication in this journal. Additionally, I have a few minor suggestions:

The content in lines 129-131 has already been mentioned earlier in the text and does not seem to add significant value here. I suggest either deleting or replacing it.

Regarding the temperature correction equation mentioned in lines 156-162, it would be beneficial to provide literature support for the basis of this method.

If I understand correctly, even if fur color affects the body temperature of cat corpses, its impact on decomposition time seems insignificant. What is the intended discussion in the second paragraph of the discussion section? Especially considering the mention of oviposition later on, do the references cited in this section align with the content discussed? I recommend revising to appropriately cite literature, discuss results accurately, and clearly express the author's own insights.

Author Response

Thank you to the reviewer for your thoughtful suggestions and feedback. We have made the changes requested and have listed the changes below:

The content in lines 129-131 has already been mentioned earlier in the text and does not seem to add significant value here. I suggest either deleting or replacing it.

Moved this sentence to appear earlier in the text.

Regarding the temperature correction equation mentioned in lines 156-162, it would be beneficial to provide literature support for the basis of this method.

Added literature.

If I understand correctly, even if fur color affects the body temperature of cat corpses, its impact on decomposition time seems insignificant. What is the intended discussion in the second paragraph of the discussion section?

Reworded/deleted areas of this paragraph to make discussion more concise and clear.

Especially considering the mention of oviposition later on, do the references cited in this section align with the content discussed? I recommend revising to appropriately cite literature, discuss results accurately, and clearly express the author's own insights.

Removed mentions of oviposition to provide a more clear explanation.

Thank you for your thoughtful consideration,

Kelly Bagsby

Reviewer 3 Report

Comments and Suggestions for Authors

This study focused on analyzing the unique decomposition characteristics of cats, specifically looking at the rate of decomposition using total body score (TBS) and accumulated degree days (ADD). The quality of this paper is very good, and it is well-written. I have a few minor suggestions. 

1. The assessment method for TBS should also be briefly explained in the materials and methods section to facilitate readers' understanding of the scoring method used in the study.

2. Since ADD is likely to be an unfamiliar term for general readers, its definition and calculation method should be detailed in the materials and methods. It was briefly mentioned at line 163, but I am still somewhat unclear on how ADD is calculated.

3. The materials and methods should specify the total duration of the study.

4. The results should detail the temperature and humidity during the study period or at least provide a range of values.

5. There is an extra "=" in "p =< 0.0001".

6. It is suggested to add a column in the figure to indicate what the TBS score approximately is for this figure.

7. In addition to figures, I would also like to see a table listing the specific values for body temperature, TBS, and ADD. This would provide a better concept when reading the discussion of differences.

6. there was no figure 5. 

7. Is Fig 6 b also Cat 10? If yes, it should be mentioned in the (b) figure legend.

8. For the significant differences shown in Figure 6, it should be clarified whether the TBS for these are excluded from the average or also included in calculating the average. This should be explained.

Author Response

Thank you to the reviewer for your thoughtful suggestions and feedback. We have made the changes requested and have listed the changes below: 

  1. The assessment method for TBS should also be briefly explained in the materials and methods section to facilitate readers' understanding of the scoring method used in the study.

Added TBS assessment method: “Each cat was photographed, and observed to document morphological changes and quantify each region to determine the TBS. When assessing the morphological changes, three regions are scored, the head/neck, torso, and limbs, are scored independently. Using the TBS chart, each region was observed and photographed. As decomposition progressed, observable changes were recorded, and a score was assigned. The three regions are then summed resulting in the total body score.”

  1. Since ADD is likely to be an unfamiliar term for general readers, its definition and calculation method should be detailed in the materials and methods. It was briefly mentioned at line 163, but I am still somewhat unclear on how ADD is calculated.

Reworded temperature/ADD section to be more clear.

  1. The materials and methods should specify the total duration of the study.

Added total duration of the study: “This study was conducted in August 2021, for a total of twelve days, in West Lafayette, Indiana, USA using twelve domestic short haired cat carcasses.”

  1. The results should detail the temperature and humidity during the study period or at least provide a range of values.

Added temperature and humidity data: “The maximum body temperature for light and dark furred cats was 39.18℃ +/- 2.2 and 41.91℃ +/- 3.3, respectively. Minimum body temperatures for light and dark furred cats was 21.30℃ +/- 1.0 and 25.39℃ +/- 1.8, respectively. Maximum relative humidity was 82.74% and minimum was 52.17%.”

  1. There is an extra "=" in "p =< 0.0001".

Deleted extra = where appropriate.

  1. It is suggested to add a column in the figure to indicate what the TBS score approximately is for this figure.

Added TBS in figure X.

  1. In addition to figures, I would also like to see a table listing the specific values for body temperature, TBS, and ADD. This would provide a better concept when reading the discussion of differences.

Added temperature and TBS table.

  1. there was no figure 5. 

Updated figure numbers.

  1. Is Fig 6 b also Cat 10? If yes, it should be mentioned in the (b) figure legend.

Added “Cat 10” to (b) section of the figure legend.

  1. For the significant differences shown in Figure 6, it should be clarified whether the TBS for these are excluded from the average or also included in calculating the average. This should be explained.

Added “Each cat was photographed, observed to document morphological changes, and decomposition was quantified for each region to determine the TBS.” and “The mean TBS was calculated for both light and dark fur experimental groups.” to the methods section.

Thank you for your thoughtful consideration,

Kelly Bagsby

Reviewer 4 Report

Comments and Suggestions for Authors

The authors present an interesting experiment on estimating the post-mortal interval using cats.

I believe it can be published after major review. 

The following are some suggestions/hurdles:

- the introduction is too long and focused on general aspects. In my opinion, it should be shortened for the parts concerning the stages of decomposition, introducing the difficulty of PMI estimation after equilibration between cadaveric and environmental temperature. I would suggest considering a recent review: " Franceschetti, Lorenzo, et al. "Estimation of Late Postmortem Interval: Where Do We Stand? A Literature Review." Biology 12.6 (2023): 783."

- materials and methods: it is not clear why the weather data were corrected and not used those from the data logger, which are certainly more reliable. Please clarify. 

- In the discussion paragraph, I would like the authors to compare the results with those on human cadavers. I would suggest for this purpose: 

Franceschetti, Lorenzo, et al. "Estimation of Late Postmortem Interval: Where Do We Stand? A Literature Review." Biology 12.6 (2023): 783.

And also: 

Pittner, Stefan, et al. "The applicability of forensic time since death estimation methods for buried bodies in advanced decomposition stages." PLoS One 15.12 (2020): e0243395.

- - It would then be interesting if the authors identified one day (the eighth for example) as the "day of corpse discovery," simulating a crime scene, and calculated the PMI with the available data by introducing the calculation formula to see how far this method deviates. 

Author Response

Thank you to the reviewer for your thoughtful suggestions and feedback. We have made the changes requested and have listed the changes below:

- the introduction is too long and focused on general aspects. In my opinion, it should be shortened for the parts concerning the stages of decomposition, introducing the difficulty of PMI estimation after equilibration between cadaveric and environmental temperature. I would suggest considering a recent review: " Franceschetti, Lorenzo, et al. "Estimation of Late Postmortem Interval: Where Do We Stand? A Literature Review." Biology 12.6 (2023): 783."

Deleted sentences/sections throughout the introduction to shorten and to be more concise.

Added Franceschetti, Lorenzo, et al. "Estimation of Late Postmortem Interval: Where Do We Stand? A Literature Review." Biology 12.6 (2023): 783." To include late PMI assessment techniques.

- materials and methods: it is not clear why the weather data were corrected and not used those from the data logger, which are certainly more reliable. Please clarify. 

Added literature and reworded sections to support temperature correction.

- In the discussion paragraph, I would like the authors to compare the results with those on human cadavers. I would suggest for this purpose: 

Franceschetti, Lorenzo, et al. "Estimation of Late Postmortem Interval: Where Do We Stand? A Literature Review." Biology 12.6 (2023): 783.

And also: 

Pittner, Stefan, et al. "The applicability of forensic time since death estimation methods for buried bodies in advanced decomposition stages." PLoS One 15.12 (2020): e0243395.

The aim of this study is animal decomposition, specifically furred animals, which is not applicable to humans. The animal models in this study were not buried and the Pittner, Stefan, et al. study is not applicable here.

- - It would then be interesting if the authors identified one day (the eighth for example) as the "day of corpse discovery," simulating a crime scene, and calculated the PMI with the available data by introducing the calculation formula to see how far this method deviates. 

The aim of this study is to provide information about TBS and providing a universal equation in order to calculate PMI is inappropriate because temperature and environmental conditions affect the rate of decomposition significantly. Stated in the introduction “Many studies have used TBS to determine the PMI, and in general, there are many shortcomings and a universal equation that is applicable to all animal species is unattainable with current resources”.

Thank you for your thoughtful consideration,

Kelly Bagsby

Round 2

Reviewer 2 Report

Comments and Suggestions for Authors

Considering that the authors have implemented the requested changes and taken into account the editorial comments, I am inclined to accept them.